EMBO
Molecular Medicine

# Seeding and transgenic overexpression of alpha-synuclein triggers dendritic spine pathology in the neocortex

Sonja Blumenstock[1,2,3] (iD), Eva F Rodrigues[2], Finn Peters[2] (iD), Lidia Blazquez-Llorca[4] (iD), Felix Schmidt[1,5], Armin Giese[1] (iD) & Jochen Herms[1,2,3,*] (iD)

## Abstract

Although misfolded and aggregated α-synuclein (α-syn) is recognized in the disease progression of synucleinopathies, its role in the impairment of cortical circuitries and synaptic plasticity remains incompletely understood. We investigated how α-synuclein accumulation affects synaptic plasticity in the mouse somatosensory cortex using two distinct approaches. Long-term *in vivo* imaging of apical dendrites was performed in mice overexpressing wild-type human α-synuclein. Additionally, intracranial injection of preformed α-synuclein fibrils was performed to induce cortical α-syn pathology. We find that α-synuclein overexpressing mice show decreased spine density and abnormalities in spine dynamics in an age-dependent manner. We also provide evidence for the detrimental effects of seeded α-synuclein aggregates on dendritic architecture. We observed spine loss as well as dystrophic deformation of dendritic shafts in layer V pyramidal neurons. Our results provide a link to the pathophysiology underlying dementia associated with synucleinopathies and may enable the evaluation of potential drug candidates on dendritic spine pathology *in vivo*.

**Keywords** alpha-synuclein; dendritic spines; *in vivo* imaging; seeding; synucleinopathies

**Subject Category** Neuroscience

## Introduction

The mammalian brain contains a complex network of billions of neurons communicating through different types of synapses that can be severely and irreversibly disturbed by neurodegeneration. In synucleinopathies like Parkinson's disease (PD) or dementia with Lewy bodies (DLB), progressive neurodegeneration is linked to misfolding and intracellular aggregation of the synaptic protein α-synuclein (α-syn; Clayton & George, 1999; Lücking & Brice, 2000) and leads to motor as well as cognitive deficits and ultimately to dementia (Spillantini *et al*, 1997; Burn, 2004; Aarsland *et al*, 2008). Familial cases of PD or DLB can be caused by mutations in the α-syn gene (Polymeropoulos *et al*, 1997; Krüger *et al*, 1998; Zarranz *et al*, 2004) or overexpression of α-syn (Simón-Sánchez *et al*, 2009) and strengthen the association between protein load/aggregation and the onset of disease. Although the filamentous, cytosolic α-syn inclusions called Lewy bodies (Spillantini *et al*, 1998) are a prerequisite for the histopathological diagnosis of PD and DLB, only an imperfect correlation between the Lewy body load and severity of cognitive impairment is known (Hughes *et al*, 1992; Parkkinen *et al*, 2005). Several studies rather support synaptic failure as the predominant pathophysiological mechanism. Presynaptic neurotransmitter release has been proven to be unbalanced in PD (Nikolaus *et al*, 2009). Moreover, presynaptic α-syn oligomers and micro-aggregates are thought to be responsible for neurodegeneration as well as for dendritic spine loss observed in postmortem DLB brains (Kramer & Schulz-Schaeffer, 2007).

Over the past few years, an increasing body of evidence supports the concept of the propagation of misfolded α-syn species from neuron to neuron, where they are transported along axons and trigger the conversion of endogenous α-syn into a pathologic form in culture and *in vivo* (Luk *et al*, 2012a,b). Aspects of progressive α-syn aggregation—namely the spread along synaptically connected brain regions and the maturing of Lewy-like inclusions—can be recapitulated in model systems by exogenous introduction of *in vitro*-generated, preformed fibrils (PFFs) of α-syn, which under certain conditions is even possible in non-transgenic animals (Luk *et al*, 2012a,b; Masuda-Suzukake *et al*, 2013; Osterberg *et al*, 2015).

Dendritic spines are small protrusions from the dendritic shaft which are highly regulated and have been shown to be altered in

1   Center for Neuropathology and Prion Research, Ludwig-Maximilians University, Munich, Germany
2   German Center for Neurodegenerative Diseases (DZNE), Munich, Germany
3   Munich Cluster of Systems Neurology (SyNergy), Munich, Germany
4   Departamento de Psicobiología, Universidad Nacional de Educación a Distancia (UNED), Madrid, Spain
5   Department of Neurology, Ludwig-Maximilians University, Munich, Germany
    *Corresponding author. Tel: +49 89 4400 46427; Fax: +49 89 4400 46429; E-mail: jochen.herms@med.uni-muenchen.de

   

several neurological disorders including neurodegenerative diseases (Fiala *et al*, 2002; Penzes *et al*, 2011; Murmu *et al*, 2013). Various studies in the field of α-syn-related disorders have addressed the impairment of dendritic spines in brain areas like the striatum (McNeill *et al*, 1988; Day *et al*, 2006; Zaja-Milatovic *et al*, 2006; Finkelstein *et al*, 2016), the hippocampus (Winner *et al*, 2012), the olfactory bulb (Neuner *et al*, 2014), and naturally the substantia nigra (Patt *et al*, 1991). However, chronic effects of cortical α-syn have not been studied extensively, even though it is becoming more and more evident that the cortical involvement in the pathophysiology of PD and DLB cannot be neglected.

This study aims to extend the understanding of the structural consequences of neocortical α-syn accumulation on dendritic spines in a time-resolved manner. To address this question, we used chronic two-photon *in vivo* imaging to monitor spine dynamics over time, in young and aged transgenic mice overexpressing human wild-type α-synuclein (PDGF-h-α-syn) (Masliah, 2000). Furthermore, we used PFFs as a tool to induce cortical α-syn accumulation in mice expressing only the reporter transgene eGFP in order to study the structural consequences of α-syn seeding on dendritic architecture and spines.

Here, we report detrimental effects on dendrites and dendritic spines due to α-syn overexpression as well as seeding. Accumulation of α-syn in the neocortex triggers the loss of dendritic spines, interferes with spine dynamics, and leads to dystrophic malformation of dendritic branches. Our combination of a transgenic and a PFF-seeding mouse model provides new insights on the α-syn-dependent mechanisms that lead to dendritic spine decay in the cortex and therefore might create a better understanding of cognitive decline in synucleinopathies.

# Results

## Overexpression of wild-type human α-synuclein leads to changes in dendritic spine density and dynamics in aged mice

*In vivo* two-photon imaging was performed in three age groups of mice (3, 6, and 12 months), which were chosen according to previously published data on the progression of α-syn accumulation and its corresponding behavioral phenotype (Masliah, 2000; Rockenstein *et al*, 2002; Amschl *et al*, 2013). We consider the 3-month-old cohort as "presymptomatic", whereas the 6- and 12-month-old cohorts show a developing and fully fledged phenotype, respectively.

Previous long-term *in vivo* imaging studies have shown that spines are highly dynamic structural elements and can be classified accordingly: Persistent spines are long lasting and stable for weeks up to the entire life span, whereas transient spines are newly formed and lost within days (Grutzendler *et al*, 2002; Trachtenberg *et al*, 2002; Holtmaat *et al*, 2005). In both h-α-syn transgenic and control mice, dendritic images were analyzed for spine density and dynamics. Newly appeared and disappeared spines relative to the previous imaging session were marked as "gained" or "lost", respectively. Spines that were gained at a certain time point and persisted for < 7 days were considered as being "transient" (Fig 1A).

Our image analysis in the 6- and 12-month-old cohorts revealed that the number of spines per μm was significantly lowered in mice overexpressing h-α-syn. The ANOVA genotype main factor was significantly changed (6 M: $F_{(1,7)} = 27.11$; $P = 0.0012$ and 12 M: $F_{(1,7)} = 55.78$; $P = 0.0001$), denoting that h-α-syn mice exhibit considerably less (≈30%) dendritic spines on apical tuft dendrites but show no increase or decrease in spine density over the imaging time period relative to controls (Fig 1B).

Concerning spine dynamics, the fractions of both gained and lost spines were found to be elevated; however, only the comparison of the lost fractions reached statistical significance (6 M: $F_{(1,7)} = 5.85$; $P = 0.0461$ and 12 M: $F_{(1,7)} = 8.86$; $P = 0.0206$). As newly gained spines were disproportionally more often lost again, the fraction of transient spines showed a significant increase in both 6- and 12-month-old h-α-syn mice (6 M: $F_{(1,7)} = 7.19$; $P = 0.0315$ and 12 M: $F_{(1,7)} = 24.25$; $P = 0.0017$; Fig 1C).

As a consequence of the change in spine dynamics, the daily turnover ratios were found to be significantly elevated in h-α-syn mice (6 M: $F_{(1,7)} = 5.65$; $P = 0.0491$ and 12 M: $F_{(1,7)} = 6.93$; $P = 0.0338$; Fig 1D). In summary, we could show that spine density is significantly reduced and does not further change over the imaging interval of 35 days. Moreover, parameters of spine dynamics were profoundly changed in mice overexpressing α-syn compared to controls. The significant increase in transient spine fraction both in 6 and 12 months of age indicates that compensatory mechanisms may take place. Since there are no significant differences between 6 and 12 months of age, spine pathology does not further progress after 6 months of age but has reached a stable equilibrium.

## Young adult α-synuclein overexpressing mice exhibit a different dynamic phenotype than aged mice

In order to reveal when the spine loss starts leading to reduced spine density in 6- and 12- month-old h-α-syn mice, we analyzed a 3-month-old cohort over 6 weeks. Different to the older cohort, we observed a progressive decrease in spine density, with a significant interaction between genotype and time in Bonferroni's *post hoc* test ($F_{(5,30)} = 15.18$; $P < 0.001$). Compared to controls, spine density in h-α-syn mice was reduced 12% on average at the first imaging time point (age of mice: 3 months) and 30% at the last imaging time point (age of mice: 4.5 months; Fig 2A).

Moreover, the dynamic fractions of spines show a somewhat contradictory picture compared to the 6- and 12-month-old mice. Our data revealed a significant impairment in the formation of new spines in h-α-syn mice ($F_{(1,6)} = 8.71$; $P = 0.0256$), while the fraction of lost spines was unaffected, which accounts for the progressive decrease in spine density. By implication, stable spines represent a significantly larger fraction in these mice ($F_{(1,6)} = 8.71$; $P = 0.0256$), while the fraction of transient spines is not significantly different from controls (Fig 2B). We also observed no significant change of the daily turnover ratio (TOR) between the groups at 3 months of age (Fig 2C).

Our results demonstrate that overexpression of human α-syn leads to structural alterations of dendritic spines in the cerebral cortex. The alterations develop already in young adults at the age of 3–4.5 months, significantly before behavioral or motoric phenotypes become apparent (Amschl *et al*, 2013), but reach a state of dynamic equilibrium later and do not progress further until 12 months of age.

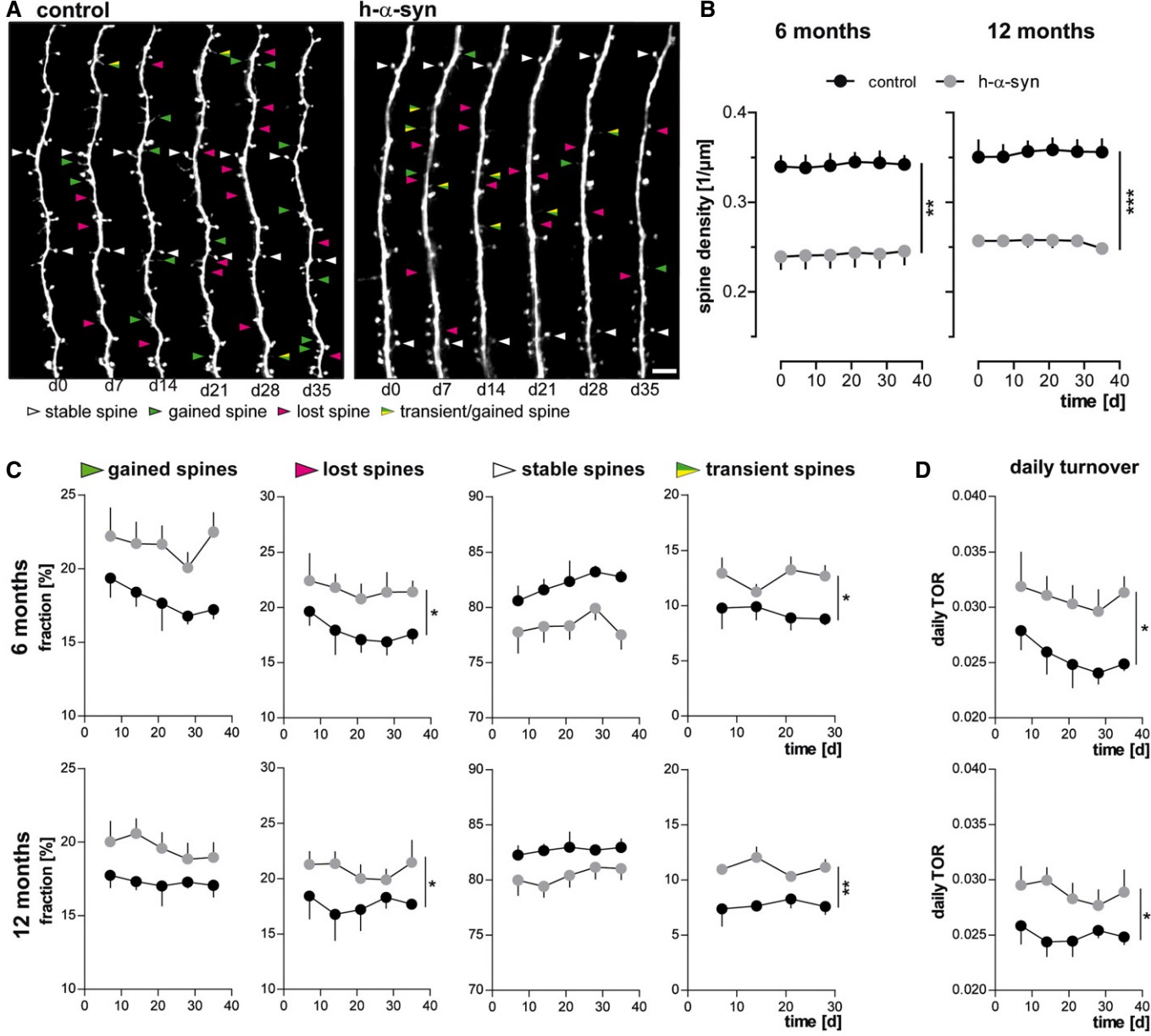

**Figure 1. α-Synuclein overexpression alters spine density and dynamics *in vivo*.**

A Representative *in vivo* two-photon recordings of eGFP-labeled apical tuft dendrites in the somatosensory cortex in h-α-syn and control animals. Arrowheads mark representative spines that were stable (white, present > 7 days), newly formed (green), or lost (magenta). Gained spines that do not stabilize (yellow/green, present < 7 days) are defined as transient. Scale bar, 5 μm.

B Spine density is reduced in both 6- (**$P$ = 0.0012) and 12-month-old (***$P$ = 0.0001) h-α-syn animals.

C The fractions of both gained ($P_{6\ months}$ = 0.0527; $P_{12\ months}$ = 0.0678) and lost spines (*$P_{6\ months}$ = 0.0461; *$P_{12\ months}$ = 0.0206) are elevated in h-α-syn mice compared to controls; the fraction of transient spines is significantly higher (*$P_{6\ months}$ = 0.0315; **$P_{12\ months}$ = 0.0017).

D Consequently, the daily turnover ratio (TOR) is significantly increased in both 6- and 12-month-old h-α-syn mice (*$P_{6\ months}$ = 0.0491; *$P_{12\ months}$ = 0.0338).

Data information: $n$ = 5 (h-α-syn), $n$ = 4 (control) animals, mean with s.e.m.; two-way ANOVA genotype main factor, *$P$ < 0.05, **$P$ < 0.01, ***$P$ < 0.001.

## Alterations in spine density and morphology in young adult and aged PDGF-h-α-syn mice

The progressive decrease in spine density seen in young adult mice could be monitored *in vivo* between 3 and 4.5 months of age. Extrapolation of the data curve (Fig 2A) of α-syn overexpressing animals to an earlier age would yield a time point at which no difference in synaptic spine densities between the groups is present. As chronic *in vivo* imaging in younger mice is technically not possible (bone growth in mice until 6–8 weeks of age, 4 weeks postsurgery necessary for chronic window clearing/healing), we addressed this question with a single time point *ex vivo* approach. Confocal

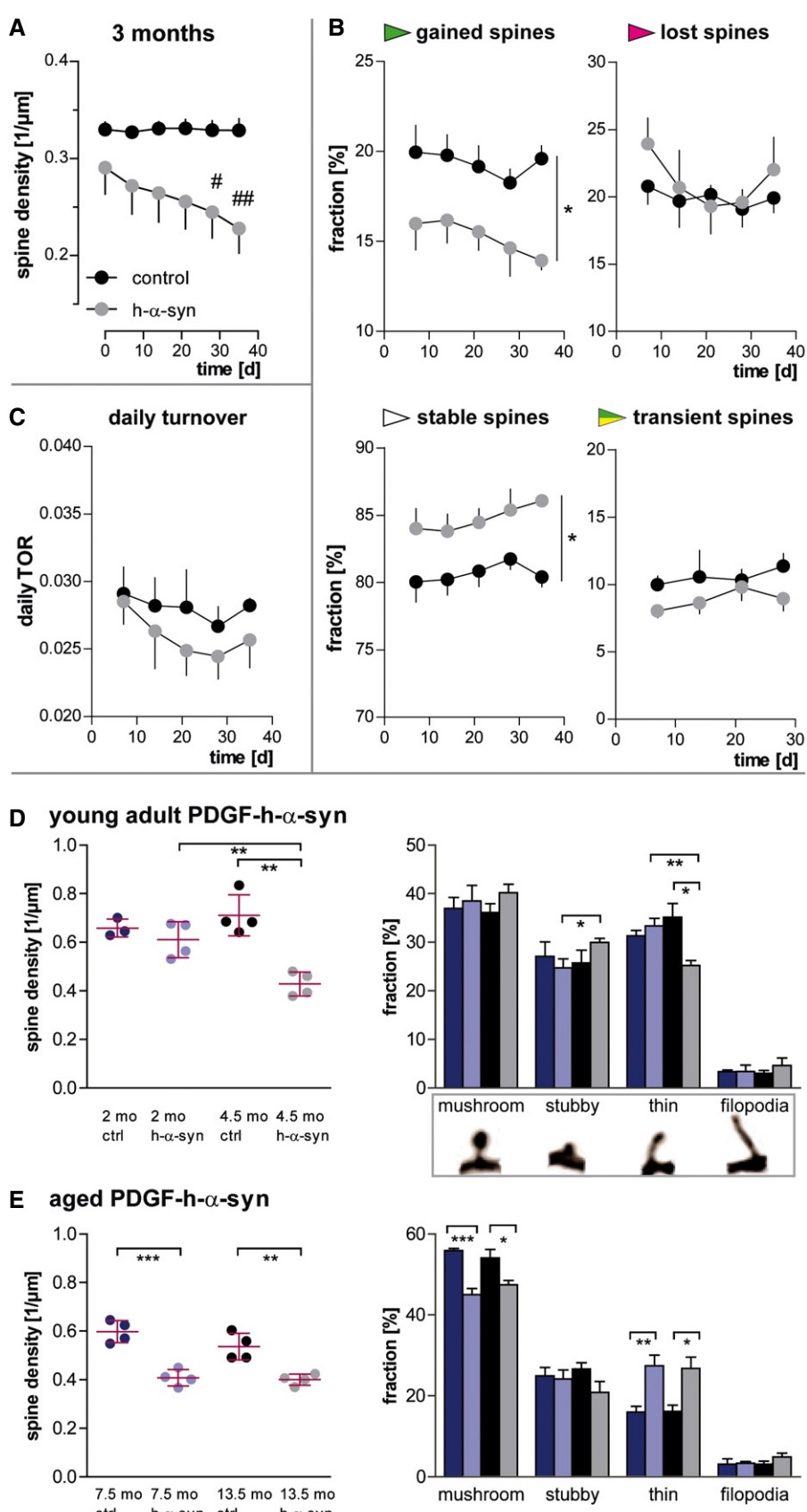

Figure 2.

◄

**Figure 2.  α-Synuclein overexpression alters spine density, dynamics, and morphology differently depending on age.**

A    In young, 3-month-old h-α-syn mice, progressive decrease in spine density is observed *in vivo*.
B    The loss of spines in young h-α-syn animals is driven by a reduced fraction of newly gained spines, while the fraction of lost spines remains unchanged and the fraction of stable spines is increased (*$P_{gained/stable}$ = 0.0256).
C    The daily turnover of spines shows no significant difference between groups ($P$ = 0.4062).
D    *Ex vivo* confocal data in young mice confirm synapse loss in h-α-syn mice between 2 and 4.5 months of age (**$P_{syn\ 2/4.5\ months}$ = 0.0064; **$P_{ctrl/syn\ 4.5\ months}$ = 0.0012) and show a shift in spine morphology toward relatively more stubby (*$p_{stubby}$ = 0.0418) and less thin spines (**$P_{h-α-syn\ 2/4.5\ months}$ = 0.0044; *$P_{ctrl/syn\ 4.5\ months}$ = 0.0155).
E    *Ex vivo* confocal data in aged h-α-syn mice show a decrease in total spine density (***$P_{7.5\ months}$ = 0.0005; **$P_{13.5\ months}$ = 0.0039) as well as in the fraction of mushroom spines (***$P_{7.5\ months}$ = 0.0004; *$P_{13.5\ months}$ = 0.0298), whereas the fraction of thin spines is increased (**$P_{7.5\ months}$ = 0.0097; *$P_{13.5\ months}$ = 0.0155).

Data information: (A–C) $n$ = 4 animals per group; (D, E) $n$ = 3–4 animals per group as illustrated; mean with s.e.m.; Bonferroni's *post hoc* test (A): #$P$ < 0.05, ##$P$ < 0.01; two-way ANOVA genotype main factor (B, C): *$P$ < 0.05, **$P$ < 0.01, ***$P$ < 0.001; Student's *t*-test (D, E): *$P$ < 0.05, **$P$ < 0.01, ***$P$ < 0.001.

imaging of apical tuft dendrites (layer I) was performed in brain slices of control as well as h-α-syn mice that were 2 and 4.5 months old. Note that the higher resolution of confocal microscopy in brain slices leads to an overall increased detection of spine density and a better ability to judge spine morphology. Two remarkable differences concerning the age and the genotype of the animal were observed. Firstly, at 2 months of age, there was no difference in spine density between control and transgenic animals, whereas at 4.5 months, spine density in h-α-syn mice showed a ~40% decrease ($t_{(6)}$ = 5.770;  $P$ = 0.0012) relative to age-matched controls. Secondly, there was a significant difference in the spine densities of the h-α-syn mice ($t_{(6)}$ = 4.097;  $P$ = 0.0064; ~30% reduction at 4.5 months) depending on their age, whereas this had no impact on the spine densities of control animals (Fig 2D).

In order to clarify whether the progressive spine loss in h-α-syn mice is reflected by changes in spine morphology, we furthermore analyzed spine shape for which three morphological subtypes have been defined (Rochefort & Konnerth, 2012). Thin spines have an elongated appearance, with a long neck and a small spine head. Mushroom spines have the largest, round head, while stubby spines are devoid of a neck and appear as round elevation from the dendritic shaft. In addition, filopodia are very long and thin structures that lack a spine head and are considered the most dynamic of dendritic structures, which appear and disappear rapidly and only sometimes develop into one of the other spine types. In the morphological analysis of our confocal data, the absolute numbers (Appendix Fig S1A) as well as the fractions of mushroom, stubby and thin spines as well as filopodia, were comparable in 2-month-old mice independent of the expression of human α-syn (Fig 2D).

In 4.5-month-old h-α-syn mice, the density of all morphological spine types is decreased (Appendix Fig S1A); however, not all types are lost to the same extent. The normalized fractions reveal a slight increase in the fraction of stubby spines ($t_{(6)}$ = 2.579; $P$ = 0.0418), whereas thin spines clearly represent a smaller fraction compared both to age-matched controls ($t_{(6)}$ = 3.346; $P$ = 0.0155) and to younger h-α-syn mice ($t_{(6)}$ = 4.443; $P$ = 0.0044) (Fig 2D).

The confocal imaging data from aged mice confirmed the reduction in spine density of ~30% in both 7.5- and 13.5-month-old h-α-syn brains. Compared to controls, the fractions of mushroom spines were decreased, while the fractions of thin spines were increased (Fig 2E and Appendix Fig S1B).

Taken together, our data confirmed our hypothesis that structural changes of dendritic spines develop within a certain time frame in young α-syn overexpressing mice. This time frame was determined to be approximately between 2 and 4.5 months of age.

Before, the dendritic appearance in these mice is indistinguishable from healthy controls. Later, α-syn overexpression causes changes in dendritic spines toward an overall lower spine density and a shift in spine morphology.

## Intrastriatal injection of preformed α-synuclein fibrils triggers a progressive spread of endogenous protein aggregation

α-syn pathology can be induced in wild-type mice by injecting preformed fibrils (PFFs) (Luk *et al*, 2012a). Using this as a tool to investigate whether seeded pathology of disease-associated α-syn will lead to structural alterations in synapses, we injected healthy 2-month-old GFP-M mice with PFFs prepared *in vitro*. Prior to injection, the quality of the injection material was controlled. The amyloid nature of PFFs was confirmed in a thioflavin T (ThT) assay, as ThT changes its fluorescent properties upon binding to amyloid β-sheet structures (Fig 3A). Differential centrifugation through a sucrose gradient confirmed the strongly increased sedimentation coefficient and thus the increased molecular density of PFFs compared to α-syn monomers (Fig 3B). Finally, an electron micrograph visualized the fibrillary structure of PFFs (Fig 3C).

Sonicated PFFs were stereotactically injected into the striatum (Fig 3D), an area which contains extensive efferent and afferent projections to other brain regions and is affected in PD (Shepherd, 2013). As cortical neurons are the targets of interest in this study, the time-course and extent of α-syn pathology progression into different layers of the neocortex was monitored at different days postinjection (dpi). The phosphorylation of α-syn at S129 (pS129) is widely considered as a pathologic posttranslational modification (Tenreiro *et al*, 2014; Oueslati, 2016) and was used in this study to identify pathologic inclusions.

Striatal injection of PFFs resulted in intraneuronal aggregation of phosphorylated α-syn (pS129-immunopositive) in the neocortex, at first confined to layer IV and V neurons (30–150 dpi), developing later to spread over all cortical layers (9 months postinjection; Fig 3E). pS129-α-syn inclusions appeared as cytoplasmic fibrillary forms (Fig 3F), which could also be stained by anti-ubiquitin (Fig 3G) and thioflavin S (Fig 3H and I), thereby presenting Lewy-body-like traits. Inclusions were also observed in the contralateral cortex relative to the injection site and the amygdala (Appendix Fig S2B). Due to a reported cross-reactivity between the pS129 antibody and phosphorylated neurofilament subunit L (NFL; Sacino *et al*, 2014b), we confirmed the presence of phosphorylated α-synuclein inclusions with a costaining for NFL (Appendix Fig S2B and C). The

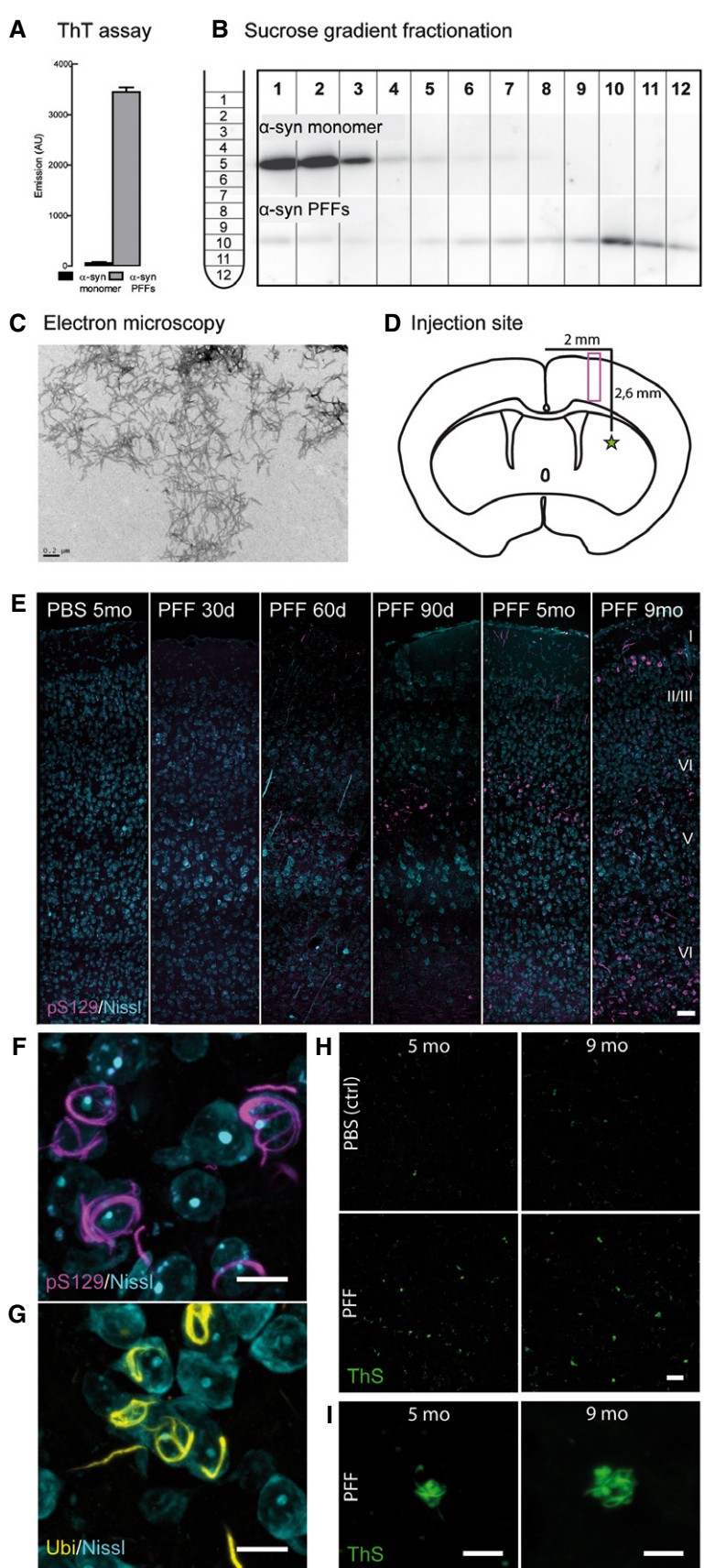

**Figure 3.**

Figure 3.    Injection of PFFs into the dorsal striatum triggers cortical α-synuclein aggregation.

A–C    The quality of the injection material was verified using ThT fluorescence assay (A), sucrose-gradient fractionation (B), and electron microscopy (C). Scale bar: 0.2 μm.

D    Injection site, 0.2 mm anterior of the bregma (star) and imaging area (box) as depicted in (E).

E–I    Representative images of the cortical layers I–VI in the somatosensory cortex of mice at different time points postinjection. Controls were injected with sterile PBS. Neurons in the layers IV and V of the somatosensory cortex contain aggregates of α-synuclein phosphorylated at S129 (F), which are ubiquitin-positive (G) as well as thioflavin S-positive (H, I). Image stacks (E–I) are depicted as maximum intensity projections. Scale bars: 50 μm (E), 10 μm (F, G), 20 μm (H), 5 μm (I).

presence of pS129-α-syn inclusions across different brain regions is shown in Appendix Fig S2D.

Control brains injected with PBS did not show comparable immunopositive inclusions (Fig 3E). Importantly, also monomeric α-syn injected into GFP-M mice as well as PFFs injected into α-syn knock-out mice (SNCA KO) failed to induce the phenotype as observed after injection of PFFs (Appendix Fig S2E).

In order to further examine whether the effect can be attributed to templated conversion of endogenous mouse α-syn into misfolded and aggregated species rather than just the uptake of PFFs by cells, control injections of fluorescently labeled PFFs into the striatum were performed. Twenty-four hours after the injection, the injection material was prominently detectable in the striatum, corpus callosum, and the injection canal (Appendix Fig S3B and C). At 6 dpi, however, the fluorescent material was removed from the injection site (Appendix Fig S3D) and later showed very little colocalization with developing pS129-α-syn-positive aggregates at 30 dpi (Appendix Fig S3E and F). We therefore conclude that the spreading of intracellular α-syn aggregation in our model is dependent on the introduction of fibrillary α-syn as a molecular template on top of endogenous α-syn expression and not simply due to a cellular uptake of injected PFFs.

Considering inflammation as a contributor in neurodegenerative diseases (Perry et al, 2010) and synaptic plasticity (Morris et al, 2013) and as PFF-induced α-syn accumulations have been associated with microglia (Sacino et al, 2014a), we assessed the presence and activation of microglia in the neocortex with two markers. Anti-Iba-1 marks all microglial cells including their processes, CD68 is a marker associated with active phagocytosis. We find that at the time point of spine analysis (5 months postinjection), the cortical area covered by microglia shows a tendency toward higher coverage, especially in layer IV (Appendix Fig S4A and B). We also observed a tendency for an increase in the coverage with activated microglia (Appendix Fig S4C and D) as well as occasional colocalization of CD8-positive microglia and pS129-positive aggregates (Appendix Fig S4E). However, variation between samples is high and the effect is not significant.

## Seeded α-synuclein aggregates have detrimental effects on dendrites and spines

Advantageous to other mouse models, the seeding model does not require a disease-related transgene to induce a pathologic phenotype. In this way, we wanted to investigate structural alterations that might be the consequence of dose- and time-controlled molecular templating of α-syn aggregates. Five months after the injection of PFFs or PBS (control) into the dorsal striatum, brain slices were stained for pS129-positive α-syn aggregates, which were most abundant in layer IV and upper layer V of the cortex, including pyramidal

eGFP-expressing neurons (Appendix Fig S3A). In order to study whether structural changes might be solely attributed to a local effect of intracellular aggregates on neighboring dendrites or due to a more global disturbance in the synaptic network, we imaged apical dendrites located in two different layers (Fig 4A). Dendrites imaged in layer IV or V were spatially close to cells bearing an intraneuronal α-syn accumulation, while tuft dendrites of layer I were distal to these cells (Fig 4B).

Indeed, we found alterations in spine density irrespective of the cortical location of the dendrites. In both layer IV/V ($t_{(11)} = 3.628$; $P = 0.004$) and layer I ($t_{(11)} = 3.838$; $P = 0.0028$), spine density was found to be lowered in brains which had been injected with PFFs and contained α-syn aggregates (Fig 4C).

We next investigated how vulnerable the morphological spine types are in response to seeded α-syn. In the layer IV/V dendrites, all types were lost to the same extent, leaving the fractions of mushroom, stubby, and thin spines comparable in controls and seeded animals (Appendix Fig S5B and Fig 4C). Contrarily, in apical tuft dendrites (layer I), thin spines proved to be more and stubby spines less vulnerable to seeded α-syn. The fraction of thin spines was therefore decreased ($t_{(11)} = 2.350$; $P = 0.0385$), while the fraction of stubby spines was increased ($t_{(11)} = 2.599$; $P = 0.0247$) (Appendix Fig S5A and Fig 4C). This phenotype was already observed in our ex vivo spine data of young transgenic h-α-syn mice (see Fig 2) and could be reproduced here. On the presynaptic side, no overt change in the density of excitatory boutons could be detected in both the transgenic and the seeding mouse model (Appendix Fig S6I–L).

The irregular form of the tuft dendritic shafts in PFF-seeded brains was an additional observation. In controls, injected only with PBS, the diameter of dendritic shafts was found to be relatively uniform. In contrast, apical tuft dendrites of PFF-seeded mice showed an irregular appearance of the dendritic shaft. Over the length of the dendrites, large swellings could be observed as well as segments with an extremely narrow diameter (Fig 4D). In a histogram plotting the frequency of diameter values, dendritic shaft diameters follow a normal distribution ($R^2 = 0.9263$) for the control animals. For the seeded mice, the variety of values is broader and the values are best described with two distributions: one at very low dendritic diameter ($R^2 = 0.9250$) and one comparable to the control distribution ($R^2 = 0.9664$), with the exception of the presence of abnormally high values (Fig 4E). As a consequence of the high variation, the mean standard deviation of the dendritic shaft diameter was found to be significantly increased ($t_{(11)} = 3.054$; $P = 0.011$) in mice that were seeded with α-syn PFFs (Fig 4F). Note that this phenotype was only observed in our seeding, but not in the transgenic model.

With our approach, we could show that the introduction of α-synuclein seeds and the consecutive accumulation in the cerebral

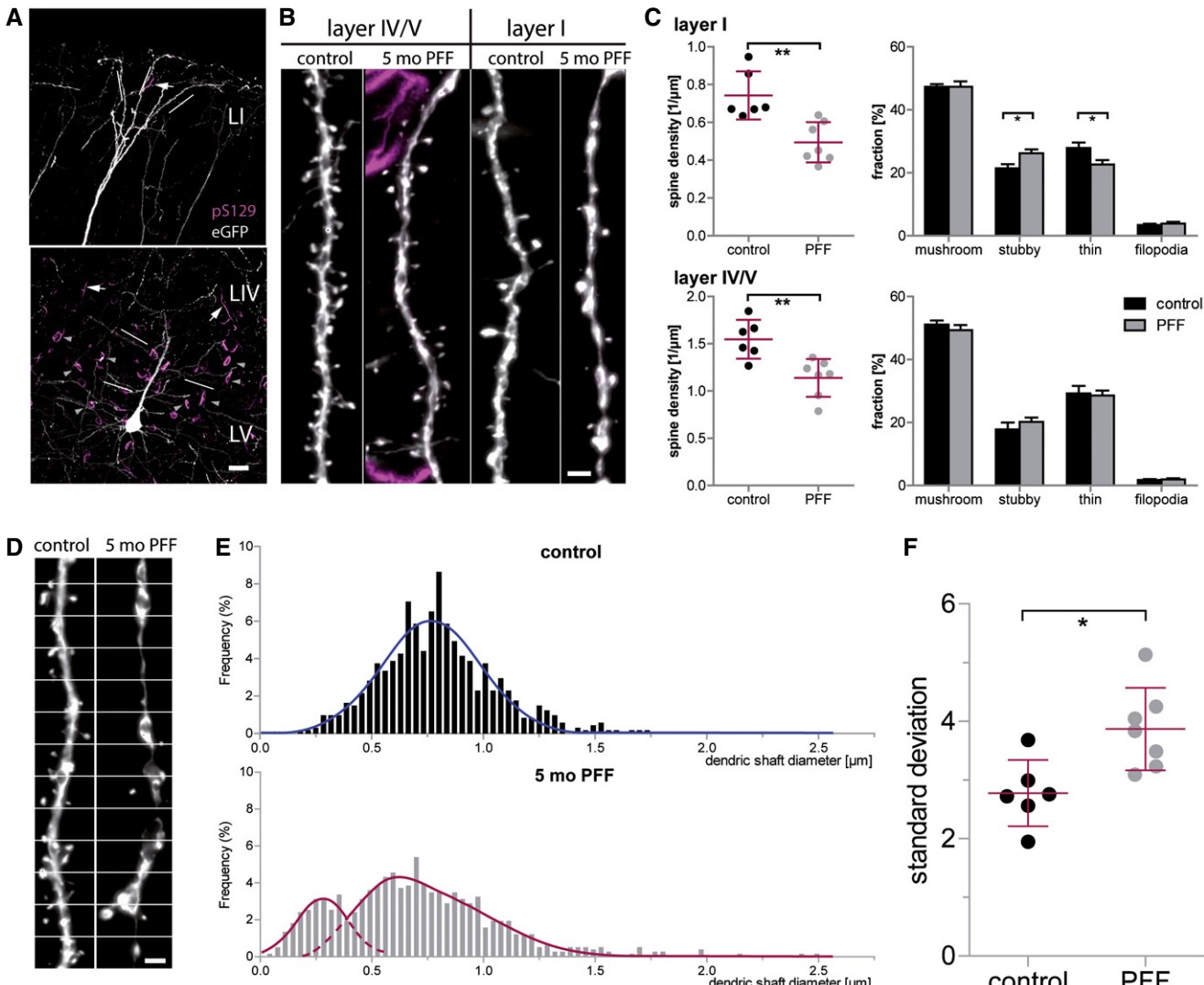

**Figure 4. The presence of accumulated α-synuclein induced by seeding 5 months prior to analysis causes spine loss and malformation in layer V apical dendrites.**

A   α-Synuclein aggregates occur as intrasomal (arrowheads) and neuritic (arrows) accumulations and are present predominantly in upper layer V and layer IV of the cortex. In layer I, pS129-positive structures are much less dense. Lines exemplarily mark dendrites used for analysis.

B   Spine analysis was performed on apical dendrites located in the cortical layers IV/V and I.

C   In layer I, spine density is reduced relative to PBS-injected controls (**P = 0.0028), with the fraction of stubby spines being increased (*P = 0.0247) and the fraction of thin spines being decreased (*P = 0.0385). In layer IV/V, spine density is reduced as well (**P = 0.004), without a significant effect on spine morphology.

D   In apical tuft dendrites of PFF-injected mice, dendrites display dystrophic swellings and parts of very small diameter; white lines: measurement positions.

E   Histograms of the dendritic shaft diameter.

F   Variation in the diameter of single dendrites in seeded mice compared to controls (*P = 0.011).

Data information: n = 6 (control), n = 7 (PFF) animals, mean with s.e.m.; Student's t-test: *P < 0.05, **P < 0.01. Scale bars: 20 μm (A), 2 μm (B, D).

---

cortex triggers structural changes both at the level of the dendritic spine and the dendritic shaft.

## Discussion

This study presents converging evidence from two distinct approaches for the adverse effects of α-syn accumulation for the density and structural plasticity of dendritic spines in the cerebral

cortex. Dendritic spines are excitatory postsynapses, and their loss has been described repeatedly as a structural correlate for cognitive impairment and dementia (Terry *et al*, 1991; Dickson *et al*, 1995; Bellucci *et al*, 2012; Picconi *et al*, 2012). Moreover, the effects of mutated and accumulated α-syn have been studied in several brain regions. Striatal spine loss has been reported in various animal models of PD and in PD and DLB patients (McNeill *et al*, 1988; Zaja-Milatovic *et al*, 2006). In a mouse model overexpressing the human A53T α-syn mutation, spine loss has been observed in the

striatum and in the hippocampus (Finkelstein et al, 2016). Some authors report that D2R striatopallidal neurons in reserpine and 6-OHDA-treated mice selectively lose spines (Day et al, 2006), while others found spine loss on both direct and indirect pathway neurons in 6-OHDA- or MPTP-treated models (Suárez et al, 2014). In contrast, another study failed to detect striatal spine loss in A53T mice, despite a decrease in the number of SNpc neurons (Oaks et al, 2013). α-syn was also demonstrated to have negative impact on newly generated neurons of the hippocampus (Winner et al, 2012) and the olfactory bulb (Neuner et al, 2014), particularly on their dendrite outgrowth and spine development.

In our current study, we aimed to extend this knowledge with a chronic examination of spines in the cortex. As the chronic effects of α-syn on cortical spine plasticity remain incompletely understood, we used two different mouse models which exhibit α-syn accumulation in the cortex. In our transgenic model, wild-type human α-syn was expressed under the PDGF promoter, leading to neocortical accumulation of mostly soluble protein located as inclusions in neuronal cell bodies as well as synaptic terminals (Masliah, 2000; Rockenstein et al, 2002; Amschl et al, 2013). Our seeding model was based on the striatal injection of preformed wild-type mouse α-syn fibrils and caused a slowly progressing phenotype of phosphorylated α-syn-positive neuronal inclusions, which first developed in the deeper layers of the neocortex and condensed over time to mature into fibrillary Lewy-like inclusions.

Our study also provides in vivo evidence that accumulation of wild-type α-syn interferes with cortical spine plasticity and spine density which declined in young adult mice starting from about 3 months of age and stabilized at a reduced overall density in older mice ($\approx 30\%$). This spine loss occurs well before behavioral impairments become apparent in these mice (Masliah, 2000; Amschl et al, 2013) and may be the result of a presymptomatic change in pre- and postsynaptic function.

While numerous studies have made great effort to unravel the molecular mechanism behind α-syn toxicity, it remains to be clarified whether a toxic gain-of-function or a loss-of-function or both is responsible for synapse and cell loss. Among the toxic gain-of-function effects of α-syn are the impairment of lysosomal and proteasomal protein degradation, induction of endoplasmic reticulum (ER) stress, Golgi fragmentation, and active formation of pores on cellular membranes (reviewed by Cookson & van der Brug, 2008). For instance, it was shown that the acute application of α-synuclein oligomers onto autaptic hippocampal cultures as well as onto hippocampal slices significantly enhances basal NMDA receptor activation and the amplitude of AMPA-receptor-mediated synaptic currents (Hüls et al, 2011; Diogenes et al, 2012). This augmented excitatory transmission could be explained by (i) α-syn oligomer-mediated pore formation in the postsynaptic membrane (Volles et al, 2001; Furukawa et al, 2006; Schmidt et al, 2012), (ii) enhancement of voltage-operated $Ca^{2+}$ channel activity (Hettiarachchi et al, 2009), or (iii) inefficient membrane repolarization (Shrivastava et al, 2015), all causing increased calcium influx which in turn results in AMPA-receptor recruitment to the dendritic spine membrane (Byth, 2014). The resulting postsynapse is saturated with receptors and therefore unable to recruit extra AMPA receptors upon theta-burst stimulation which causes a long-term potentiation (LTP) decline and overall excitotoxicity (Yuste & Bonhoeffer, 2001; Diogenes et al, 2012). On the long term—as we

analyzed the structural impact of transgenic and seeded α-synuclein during and after several months—this proposed mechanism based on a slightly but chronically enhanced calcium influx may disturb the tightly controlled, calcium-dependent machinery of synaptic plasticity. Likewise, a primary presynaptic pathology might alter synaptic vesicle release which as a long-term consequence may alter dendritic spine plasticity. Excessive α-synuclein has been shown to inhibit neurotransmitter release via specifically reducing the size of the synaptic vesicle recycling pool (Nemani et al, 2010) as well as to regulate SNARE-driven membrane fusion (Garcia-Reitbock et al, 2010; Burre et al, 2015). A study in DLB brain tissue showed massive deposits of diffuse aggregates in the cortical and subcortical gray matter, which were located at synaptic terminals and were linked to dendritic spine loss. This suggests that the subcellular presynaptic location of α-syn is important for its neurodegenerative effect (Kramer & Schulz-Schaeffer, 2007) and although our models show no overt presynaptic loss, an alteration in function seems likely. The functional roles of α-synuclein, in conjunction with regulating actin dynamics and chaperoning the polymerization of microtubule-associated proteins (Sousa et al, 2009), would certainly impair spine stability and dendritic plasticity (Tsaneva-Atanasova et al, 2009).

The α-syn loss-of-function hypothesis comprises that with progressing aggregation, endogenous α-syn becomes sequestered into inclusions; be they oligomers, protofibrils, or mature LBs and LNs, leaving less functional protein to perform normal cellular functions (reviewed by Benskey et al, 2016). Particularly the changes seen after seeding with PFFs are supportive of this hypothesis. As previously shown and confirmed in this study through seeding of fluorescently tagged α-syn fibrils, the aggregates formed are chiefly composed of endogenous α-syn, with very little of the initial PFF seed being part of the developed inclusions (Luk et al, 2009). The concept of sequestration is further reinforced by the complete absence of inclusions and toxicity in α-syn null mice seeded with PFFs (Volpicelli-Daley et al, 2011; Luk et al, 2012a). To complicate things even further, recent evidence suggests that microglial activity can contribute to both detrimental and protective mechanisms in neurodegenerative diseases (Morris et al, 2013; Miyamoto et al, 2016; Tang & Le, 2016). In our model, the infusion of PFFs did not cause overt and chronic microglial activation at the observed time point, but additional experiments might be necessary to gain a broader picture about the glial involvement in α-syn-affected synapse dynamics.

From our imaging data, we cannot pin down which of these described mechanisms exactly are involved in the loss of spines, which emphasizes the need for further studies for clarification. Furthermore, as our in vivo data suggest, the mechanism underlying synapse loss and the shift in morphology may not even be uniform throughout the course of disease. Spine dynamics were found to be changed remarkably depending on the age of the animal upon observation. The decrease in gained spines observed in 3-month-old animals caused a progressive spine loss and is morphologically reflected in the diminished fraction of thin spines, which are known to correspond to non-synaptic transient precursors of the larger stabilized spines (Holtmaat et al, 2005; Arellano et al, 2007). Contrarily, in the two older groups (6 and 12 months), further spine loss was restrained by an elevation of the daily turnover ratios. This was in turn reflected by an increase in the fraction of thin spines.

Thus, we propose that the observed spine remodeling in these animals represents a compensatory mechanism for the loss of absolute spine number, thereby strengthening the remaining synaptic contacts.

In both our seeding and transgenic model, proteinaceous aggregates are present in the neocortex, albeit in two histopathological variances: while the PDGF promoter in the transgenic model drives a more or less uniform accumulation of non-fibrillary (Masliah, 2000) α-syn in cortical neurons and synapses, seeding of PFFs leads to the formation and maturation of fibrillary Lewy-like structures in neurons connected to the dorsal striatum. Our results from our *in vivo* study in young transgenic mice are in line with the dendritic spine phenotype which can be induced through inoculation with PFFs. Dendritic spine loss in layer V pyramidal neurons as well as a redistribution of spine morphologies toward less thin spines can be observed 5 months after inoculation with PFFs. The observed spine loss was independent of whether the respective dendrite was spatially close to the cortical layer V, in which neuronal α-syn

inclusions were most pronounced. Interestingly, dystrophic deformation of apical tuft dendrites was present in PFF-seeded brains. Dendritic varicosities have been described in other progressive neurodegenerative diseases and have been linked to dendritic spine loss in scrapie-injected rodents (Fuhrmann *et al*, 2007) and an Alzheimer's disease model (Bittner *et al*, 2010). The presence of axonal varicosities caused by α-synuclein seeding has been reported (Rockenstein *et al*, 2005); furthermore, it has been shown that inclusions mature and condense over time and that aggregate bearing neurons selectively degenerate (Osterberg *et al*, 2015). Therefore, it seems likely that the degeneration of the respective cell body and the accompanying transport deficits are the cause for dystrophies and contribute to spine loss in our PFF-seeded brains.

For answering scientific questions, animal models in principle comprise advantages as well as drawbacks (Bezard *et al*, 2013). For example, all available genetic mouse models of synucleinopathies are able to model only a subset of symptoms known in the human disease cases. Promoter-driven protein expression is

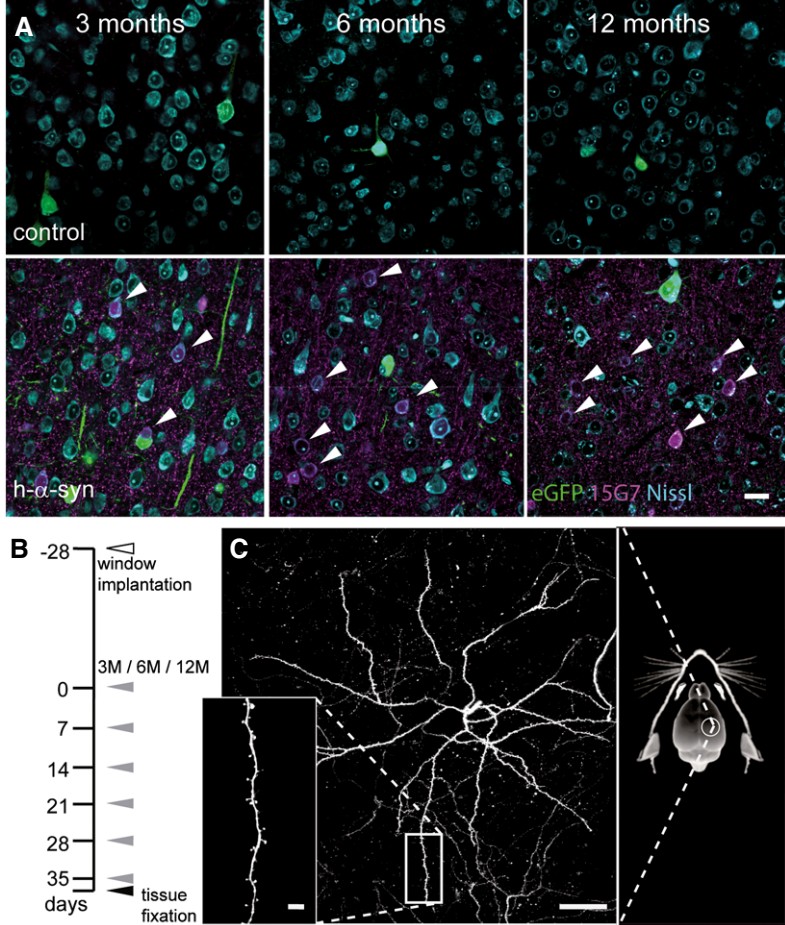

**Figure 5. Cortical α-synuclein accumulation and long-term *in vivo* imaging of PDGF-h-α-syn × GFP-M mice.**

A Immunostaining with 15G7 antibody shows cortical layer V overexpression of α-synuclein (magenta) including accumulation in cell bodies (arrows).

B Experimental timeline: 4 weeks after window implantation (white arrowhead), imaging was performed over 6 weeks, once every 7 days (gray arrowheads) and finally followed by perfusion and tissue fixation directly after the last imaging session (black arrowhead).

C Overview and detailed projections of eGFP-labeled layer V apical dendrites in the somatosensory cortex, imaged through a chronic cranial window.

Data information: Scale bars: 20 μm (A), 50 μm (C), 5 μm (C, inset).

 

limited to a cell population in which the promoter is active and is even complicated through the fact that background strain might affect the expression pattern (Chesselet *et al*, 2008). Apart from that, α-synuclein pathology induced by seeding of PFFs can be induced in wild-type animals (Luk *et al*, 2012a) in a dose- and region-specific manner. Here, however, the unphysiologically large protein amount used for "planting the seed" should be considered. One strong point of our combined approach was to apply and compare both models, in order to best break down the actual structural impact of α-syn accumulation on cortical neurons. Collectively, our results support the idea that α-syn homeostasis is constantly balanced on a knife-edge and that perturbations that change its expression produces neurotoxicity with a common mechanism in which the critical events are protein misfolding, aggregate deposition, aggregate propagation, and synaptic dysfunction (Jucker & Walker, 2011; Guo & Lee, 2014; Luna & Luk, 2015). Our data provide evidence for the harmful structural consequences of excessive wild-type α-synuclein on cortical dendrites throughout different pathological stages and may pave the way toward new therapeutic strategies for PD and DLB that focus on the maintenance of synaptic function.

# Materials and Methods

### Animals

PDGF-h-α-syn animals were obtained from QPS Austria Neuropharmacology (Grambach, Austria) and bred on a C57Bl/6 background. These transgenic mice overexpress human α-synuclein (h-α-syn) under the regulatory control of the human platelet-derived growth factor-β (PDGF-β) promoter (Masliah, 2000). Expression is strongest in the neocortex, hippocampus, olfactory bulb, and limbic system, leading to intraneuronal and synaptic α-syn accumulation as early as 3 months of age (Fig 5A and Appendix Fig S6A–D) as well as progressive motor deficits (Masliah, 2000; Amschl *et al*, 2013). Furthermore, C57Bl/6 wild type, Thy1-eGFP transgenic mice (GFP-M) (Jackson Laboratory, Bar Harbor, ME, USA), and α-syn knock-out (SNCA KO) mice (Abeliovich *et al*, 2000) were used. For *in vivo* imaging, the heterozygous PDGF-h-α-syn × GFP-M (termed h-α-syn in this paper) line was created by interbreeding and the experiment involved three age cohorts of both male and female mice, in which mice were 3, 6 or 12 months old at the first imaging time point. All animals were housed in groups under pathogen-free conditions and bred in the animal housing facility at the Center for Neuropathology and Prion Research of the Ludwig Maximilians University of Munich, with food and water provided *ad libitum* (21 ± 2°C, at 12/12-h light/dark cycle). After cranial window implantation, mice were housed separately. All experiments were approved by the Bavarian government (Az. 55.2-1-54-2532-163-13) and performed according to the animal protection law.

### Injection material and stereotactic injection

Recombinant wild-type mouse α-synuclein was expressed in BL21 (DE3) *E. coli* using a pRK172 plasmid (kind gift from Kelvin Luk and Virginia Lee, University of Pennsylvania, USA) as previously described (Nuscher *et al*, 2004; Kostka *et al*, 2008). Briefly,

*Escherichia coli* BL21(DE3) (Invitrogen, MA, USA) were transformed with the plasmid, and expression was induced with isopropyl β-D-1-thiogalactopyranoside (IPTG, Peqlab, Erlangen, Germany). Cells were lysed by boiling after heat-inactivation of proteases. After centrifugation, the supernatant was filtered through Filtropur S 0.2 filters (Sarstedt, Nümbrecht, Germany), loaded on a HiTrap Q HP anion-exchange column (5 ml, GE Healthcare, Munich, Germany) and eluted with a linear 25 mM to 500 mM NaCl gradient. Synuclein containing fractions were concentrated using VivaSpin 2 columns (Sartorius, Göttingen, Germany). Protein concentration was assessed to 5 mg/ml in 50 mM Tris–HCl, pH = 7.0. After freezing in liquid nitrogen, the protein was stored at −80°C. Appendix Fig S2A shows blots from the monomer preparation. Preformed fibrils (PFFs) were assembled from purified α-synuclein monomer (5 mg/ml) by incubation at 37°C with constant agitation (1,400 rpm) in an orbital mixer (Eppendorf, Hamburg, Germany) for 96 h and stored at −80°C (Conway *et al*, 2000; Deeg *et al*, 2015). For fluorescent labeling, purified α-synuclein monomer (3.75 mg/ml, containing 100 mM NaHCO$_3$) was incubated with 0.34 mg/ml Alexa Fluor® 488 NHS Ester (Life Technologies, Darmstadt, Germany) for 18 h at 4°C. The remaining free fluorophore was removed from the solution using PD-10 columns (GE Healthcare, PA, USA) according to the manufacturer's recommendations prior to fibril assembly as described above. Directly before injection, an aliquot of PFFs was sonicated four times with a handheld probe (SonoPuls Mini 20, MS1.5, Bandelin, Berlin, Germany) according to the following protocol: amplitude 30%; time 15 s (pulse on 3 s, pulse off 6 s). Two-month-old mice (male or female) were anesthetized with ketamine/xylazine (0.13/0.01 mg/g body weight; WDT/Bayer Health Care, Garbsen/Leverkusen, Germany) and stereotactically injected with 5 μl (25 μg) of PFFs into the dorsal striatum (coordinates relative to the bregma: +0.2 mm anterior, +2.0 mm from midline, +2.6 mm beneath the dura) of the right hemisphere using a 5-μl Hamilton syringe. Injections were performed at 400 nl/min with the needle in place after injection for at least 5 min. Control animals received sterile PBS or 5 μl of 5 mg/ml monomeric α-synuclein. Animals were monitored regularly after the surgery and sacrificed at different predetermined time points (30 days, 60 days, 90 days, 5 months, and 9 months after injections) for perfusion and tissue fixation with 4% paraformaldehyde.

### Thioflavin T binding and sucrose gradient

For *in vitro* measurement of thioflavin T (ThT) binding, 10 μM ThT was added to 100 μg/ml monomeric α-synuclein or PFFs and incubated at 30°C for 10 min. Fluorescence signal was excited at 420–460 nm and emission detected at 460–500 nm with a spectrofluorometer (FluoStar Optima, BMG Lab Tech, Jena, Germany). Continuous sucrose-gradient assay was performed as described previously (Friedlander *et al*, 2002; Wagner *et al*, 2013). Briefly, six layers of solutions with decreasing sucrose concentration (50 mM Tris pH 7.5, 0.1% NP-40, 10–60% D(+)-sucrose, respectively) were filled into a 4 ml 11 × 60 mm polyallomer tube (Beckman Coulter, CA, USA). Finally, 200 μl of 5 μM protein in 1× TBS (pH 7.5) containing 0.1% NP-40 was loaded on the top of the gradient. Ultracentrifugation with 100,000 *g* at 4°C for 1 h was performed using a Sw60Ti rotor (Beckman Coulter, USA). Resulting continuous gradients were

fractionated in volumes of 200 µl. 20 µl per fraction was analyzed by denaturing Western blot using a monoclonal antibody against mouse-α-synuclein (New England Biolabs, Frankfurt, Germany).

### Electron microscopy

α-Synuclein fibrils diluted to 5 µM in $H_2O$ were adsorbed onto a Formvar-coated, carbon-stabilized copper grid. The grid was then rinsed briefly with $H_2O$ and stained with uranyl acetate and lead citrate. Digital images were captured at different magnifications on a Jeol JEM-1011 TEM (JEOL Inc., MA, USA) equipped with an 11 megapixel Orius CCD digital camera (Gatan Inc. CA, USA).

### Cranial window

A cranial window was implanted over the right cortical hemisphere as previously reported (Fuhrmann *et al*, 2007). In short, the mice were given an intraperitoneal injection of ketamine/xylazine to reach surgical anesthesia. Additionally, dexamethasone (0.01 mg/g body weight; CP Pharma, Burgdorf, Germany) was intraperitoneally administered immediately before surgery (Holtmaat *et al*, 2009). A circular piece of the skull (4 mm in diameter) over the right hemisphere (centered over the parietal bone, approx. 5.5 mm caudal from the bregma and 5.5 mm lateral from midline) was removed using a dental drill (Schick-Technikmaster C1; Pluradent; Offenbach, Germany). The craniotomy was closed immediately with a round coverslip (4 mm in diameter), held with dental acrylic. A small *z*-shaped titan bar was glued next to the coverslip to allow repositioning of the mouse during subsequent imaging sessions. After surgery, mice received subcutaneous analgesic treatment with carprofen (5 mg/kg body weight; Rimadyl; Pfizer, NY, USA) and antibiotic treatment with cefotaxime (0.06 mg/kg body weight; Sanofi-Aventis, Frankfurt, Germany).

### Long-term *in vivo* imaging

Imaging started 4 weeks after the cranial window preparation to allow the animals to recover from surgery. After 6 weekly 2-photon *in vivo* imaging sessions, mice were sacrificed and their brains were fixed for further analyses (Fig 5B). Two-photon imaging was performed on a LSM 7 MP (Zeiss, Jena, Germany) equipped with GaAsP detectors and a 20× water-immersion objective (W Plan-Apochromat 20×/1.0 DIC, 1.0 NA, Zeiss, Jena, Germany). eGFP was excited at 880 nm by a Ti:Sa laser (MaiTai DeepSee, Spectra-Physics, Darmstadt, Germany), and emission was collected from 440 to 500 nm. Image stacks of $425 \times 425 \times 249 \ \mu m^3$ were acquired using the "*z*-stack" mode of the microscope control software (ZEN 2012/ZEN 2010 64-bit) with a lateral resolution of 0.83 and 3 µm separation distance between consecutive images. Mice were anesthetized with isoflurane (CP Pharma, Burgdorf, Germany) for imaging and fixed to a custom-made head holder using the attached metal bar. In subsequent imaging sessions, previously imaged volumes were identified by eye using the unique blood vessel pattern and fine adjusted by the position of previously imaged dendrites. This method enabled precise alignment of the same imaging volume over a period of 6 weeks (Fig 5C; Helmchen & Denk, 2005). In order to maintain stable fluorescence emission levels, the laser power was adjusted relative to imaging depth.

### Image processing and analysis of *in vivo* and *ex vivo* data

For image analysis, only microscope image data with sufficient eGFP expression and good signal-to-noise ratio over time were included and the experimenter was blinded to the genotype of the mice by assigning a random number to each animal. From each mouse (male or female, $n = 4$–5 *in vivo*, $n = 6$–7 seeding experiment, without randomization), 9–10 dendrites from at least three different positions underneath the cranial window or in slices were analyzed. The length of individual dendrites was measured in ZEN 2012 (Zeiss). Spines identified along the dendrite were marked as gained, lost, or stable. The mean density of dendritic spines was estimated for each time point and expressed over 1 µm of dendrite length. The stability of spines was calculated based on the amount of spines that remained unaltered for at least two subsequent imaging sessions. The spine turnover rate (TOR) was assessed based on gain and loss of spines over each day of imaging, calculated as follows: $TOR = (N_{gained} + N_{lost})/(2 \times N_{present})/I_t$, where $N_{gained}$, $N_{lost}$, and $N_{present}$ represent the number of gained, lost, or total spines at time points of interest, respectively, while $I_t$ is the number of days between consecutive imaging sessions. *Ex vivo* image analysis measurements were performed manually from maximal projection images of deconvoluted (AutoQuantX3, Media Cybernetics) confocal stacks. All spines along the dendrite were marked and categorized into three morphologically different classes, according to established criteria (Jung *et al*, 2011). For analysis of the dendritic diameter, length measurements were taken every 40 pixels along the dendritic shaft (control: 756 measurements, $n = 6$, PFF-seeded: 840 measurements, $n = 7$) using ImageJ. To quantify the density of glutamatergic presynaptic boutons VGLUT1-positive Imaris 7.7.2 software was applied to detect VGLUT1-positive puncta. In detail, the spots detection algorithm was applied using an estimated diameter of 0.5 µm in *xy* and an 1.5 µm in *z*. Background subtraction was enabled, and region-growing type was set to local contrast with a manual threshold of 30 defined for all datasets. Only spots with a *xy* diameter greater than 0.4 µm were counted as boutons. The data were compiled in MATLAB using ImarisXT interface.

### Statistical analysis

The sample size of animals and imaged dendrites per animal were chosen according to our previous experience in long-term imaging (Neuner *et al*, 2014; Filser *et al*, 2015). Graphs were created, and statistics were calculated in Prism v 5.04 (GraphPad Software, San Diego, CA, USA). For *in vivo* time series data, two-way ANOVA followed by the Bonferroni's *post hoc* test was used to compare the variance of spine parameters assessed over time in control and h-α-syn animals. For assessment of inter-group differences at single time points, Student's *t*-test (unpaired, two-sided) was applied. Normal distribution was assumed according to the central limit theorem, as spine densities were calculated as the means of means for every mouse. For *t*-tests, the variance between groups was tested (*F*-test) and not found to be significantly different. Data are expressed as mean ± SEM unless otherwise indicated, with $P < 0.05$ defining differences as statistically significant (*$P < 0.05$; **$P < 0.01$; ***$P < 0.001$; for *post hoc* test: #$P < 0.05$; ##$P < 0.01$).

**The paper explained**

**Problem**

Parkinson's disease (PD) and is the second most common neurodegenerative disease with a prominent loss of nigrostriatal dopaminergic neurons. The resultant dopamine deficiency underlies the onset of typical motor symptoms. Apart from this, PD and other neurodegenerative conditions featuring misfolded and aggregated forms of the synaptic protein α-synuclein (α-syn) are frequently accompanied by cognitive decline and dementia, arising from the structural and functional changes in the brain cortical synapses. The role of α-syn in the impairment of cortical circuitries and synaptic plasticity remains incompletely understood. We therefore investigated how α-synuclein accumulation affects the plasticity of dendritic spines as the loss of these small protrusions from the neuronal dendritic shaft is widely considered a structural correlate for cognitive decline.

**Results**

Our study of the cortex involved two distinct *in vivo* mouse models: Long-term *in vivo* two-photon imaging of apical dendrites was performed in mice overexpressing wild-type human α-syn. Additionally, intracranial injection of preformed α-syn fibrils was performed to seed cortical α-syn pathology in mice without an additional disease-related transgene. We find that α-synuclein overexpressing mice show decreased spine density and abnormalities in spine dynamics in an age-dependent manner. We also provide evidence for the detrimental effects of seeded α-syn aggregates on dendritic architecture, as we observed spine loss as well as dystrophic deformation of dendritic shafts in layer V pyramidal neurons of the cortex.

**Impact**

Our results demonstrate the impact of cortical α-syn in a time-resolved manner and provide a link to the pathophysiology underlying dementia associated with α-synucleinopathies. They may also enable the evaluation of potential drug candidates on dendritic spine pathology *in vivo*.

**Immunohistochemistry and confocal microscopy**

Animals in deep ketamine/xylazine anesthesia were killed by transcardial perfusion with 1× phosphate-buffered saline (PBS) followed by 4% paraformaldehyde (w/v). The brains were postfixed in PBS containing 4% paraformaldehyde overnight before cutting 60-μm-thick coronal sections on a vibratome (VT 1000S from Leica, Wetzlar, Germany). Immunofluorescence staining was performed on free-floating sections. Primary antibody incubation (overnight or 48 h (anti-VGLUT1), 4°C) was followed by 2 h of secondary antibody incubation at room temperature. Appendix Table S1 lists the antibodies used in this study. For unambiguous analysis of GFP-labeled neurons in eGFP-M mice, slices for spine analysis were re-stained with anti-GFP Alexa Fluor® 488 antibody (Invitrogen, Life Technologies GmbH). Sections for the study of neuronal α-syn accumulation were incubated with a fluorescent Nissl stain (Neurotrace™ 435/455, Thermo Fisher Scientific, USA). For mounting on glass coverslips, VECTASHIELD® Mounting Medium (Vector Laboratories) was used. Laser wavelengths used for excitation and collection range of emitted signals were as follows: Alexa Fluor® 488/eGFP—488 nm/500–550 nm; Alexa Fluor® 594—561 nm/585–743; Alexa Fluor® 647—633 nm/long-pass 650 nm; Nissl—750 nm/435–485 nm. Analysis of dendritic spines was limited to cortical apical dendrites from layer V neurons, cropped, and imaged at high

resolution. For imaging of presynaptic boutons, three-dimensional 16-bit data stacks of 1,024 × 1,024 × 26 pixels were acquired from four different positions in the somatosensory cortex at a lateral resolution of 0.1 μm/pixel and an axial resolution of 0.4 μm/pixel. Appendix Fig S6E–H illustrates the automatic bouton detection applying the algorithm.

**Expanded View** for this article is available online.

## Acknowledgements

We thank Katharina Bayer, Sarah Hanselka, and Eric Grießinger for their excellent technical support and animal care. We like to give special thanks to Prof. Virginia Lee and Dr. Kelvin Luk for providing us with the plasmid for α-syn expression and to S. Tahirovic for advice and kindly providing the CD68 antibody. We also thank MM. Dorostkar, S. Crux, S. Filser, and C. Sgobio for scientific support and advice on the manuscript. This work was funded the Munich Cluster for Systems Neurology SyNergy (EXC 1010).

## Author contributions

SB performed design, data collection, analysis and interpretation of experiments and wrote the manuscript. EFR contributed to the data collection in *in vivo* experiments. FP analyzed presynaptic terminals. LB-L provided electron micrographs from α-synuclein PFFs. AG and FS provided experimental material and expertise in α-synuclein purification in PFF assembly. EFR and FS helped with manuscript preparation. JH supervised the study; contributed to conception, design, and manuscript writing; and provided financial support and final approval of the manuscript.

## Conflict of interest

The authors declare that they have no conflict of interest.

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

  α-synuclein causes parkinson and Lewy body dementia. *Ann Neurol* 55:
  164−173

