## [Review Process File · EMBO Molecular Medicine]

Seeding and transgenic overexpression of alpha-synuclein triggers dendritic spine pathology in the neocortex

Sonja Blumenstock, Eva F. Rodrigues, Finn Peters, Lidia Blazquez-Llorca, Felix Schmidt, Armin Giese and Jochen Herms

Corresponding author: Jochen Herms, DZNE - German Center for Neurodegenerative Diseases

Review timeline:

Submission date:	07 November 2016
Editorial Decision:	13 December 2016
Revision received:	06 February 2017
Editorial Decision:	22 February 2017
Revision received:	27 February 2017
Accepted:	02 March 2017

Transaction Report:

Editor: Céline Carret

1st Editorial Decision

13 December 2016

Thank you for the submission of your manuscript to EMBO Molecular Medicine. We have now heard back from the three referees whom we asked to evaluate your manuscript. Although the referees find the study to be of potential interest, they also raise a number of concerns that need to be fully addressed in the next final version of your article.

You will see from the comments pasted below that all three referees are rather supportive of publication. Nevertheless they all make suggestions to render the study even more compelling and we would like to encourage you to address the issues raised. Of particular relevance, please thoroughly discuss the mechanisms as suggested by referees 1 and 2, and medical relevance as highlighted by referee 3.

Given the balance of these evaluations, we feel that we can consider a revision of your manuscript if you can address the issues that have been raised within the time constraints outlined below. Please note that it is EMBO Molecular Medicine policy to allow only a single round of revision and that, as acceptance or rejection of the manuscript will depend on another round of review, your responses should be as complete as possible.

Please also contact us as soon as possible if similar work is published elsewhere. If other work is

published we may not be able to extend the revision period beyond three months.

I look forward to receiving your revised manuscript.

1st Revision - authors' response

06 February 2017

Referee #1

We thank this referee for his/her encouraging comments with constructive suggestions. In the revised text we make necessary clarifications and provide additional experimental data requested along with the rebuttal of specific points made.

Specific points:

1. "For example, is the loss of spine cell-autonomous or resulted from axon denervation? It'd be interesting to examine the presynaptic changes at the same time."
 1. **As suggested, we carried out additional experiments to examine the presynaptic density in both transgenic PDGF-h- α -syn and PFF-seeded mouse brain slices. Using a staining against the vesicular glutamate transporter 1 (VGLUT1) marking excitatory presynapses, we could not find the presynaptic bouton density to be changed in neither mouse model compared to their respective controls (see new Supplementary figure S6). Due to the variability between mice, small changes in the generally high density of boutons might be elusive. Previous studies also showed that spines show a higher plasticity than boutons (Liebscher *et al*, 2014) and that transient spines can exist without a connected bouton (Knott *et al*, 2006), which might make the detection of subtle changes difficult. Furthermore, functional alterations cannot be addressed by this method, which might also be present and could be addressed in future studies.**
2. "In addition, did the infusion of alpha-synuclein cause microglia activation? In light of the role of microglia in synaptogenesis, does neuroinflammation contribute to the loss of spines?"
 2. **We thank this referee for bringing up this relevant question. As requested, we provide additional experiments regarding the activation of microglia in brain slices after the infusion of α -syn PFFs. We have stained brain slices from the same group of mice that were used for spine analysis (5 months post injection of PFFs or PBS) for two Microglia markers: CD68 and Iba1. We find that the cortical area covered by the signal from these microglial markers does not significantly differ between groups. In some PFF brains, a concentration of microglia around cortical layer IV – the region of strongest pS129 positive α -synuclein accumulation – is visible. This effect however shows great variance across individuals and therefore yields no statistical significance. These points are now highlighted in the discussion of the revised manuscript and summarized in the new supplementary figure S4 and necessary additional references were made.**

Referee #2

We thank this referee for numerous valuable comments and suggestions along with critics. We appreciate his/her interest in our work with encouraging remarks and constructive suggestions. We now provide the additional experimental data requested along with the rebuttal of specific points made.

Specific points:

1. "It is extremely important to validate alpha-synuclein expression in the experimental animals, since authors breed PDGF promoter driven alpha-synuclein mice with Thy1 promoter driven GFP mice. Figure 1 does not demonstrate convincingly that GFP-expressing dendrites would have alpha-synuclein in them, if anything it shows that there are less GFP-positive processes in 12 months old mice compared to 3 months old mice. What mice were used as controls?"
 1. **We thank this referee for bringing up this relevant question. Our staining in Figure 1 is meant to demonstrate the presence of human α -synuclein overexpressed in the somatosensory cortex. The punctuate neuropil staining pattern of α -synuclein has been described to be consistent with presynaptic terminals (Kahle, 2007), the**

physiological location of α -synuclein. In our new supplementary figure S6, we show colocalization between human α -syn and glutamatergic presynapses. Concerning the presence of α -syn in dendrites, the expression of the eGFP in dendrites used for imaging diminishes the possibility of antibody absorption for immunostaining. Independently of this, we do not claim to know that the spine loss observed in our models is due to the dendritic presence of α -synuclein, but could rather be a consequence of an altered presynaptic function and overall network activity. We plan to provide further knowledge on this in future studies.

2. “Authors suggest that changes in dendritic spines occur within 2-4.5 months of age in alpha-synuclein overexpressing mice. They provide description of morphological subtypes change in 4.5 months mice (Fig 3e), and state that "later, a-syn overexpression causes changes in dendritic spines towards...a shift in spine morphology" (p12, last sentence). This has to be supported by quantitative data, as there is no data presented showing changes in the morphology of spines at 6 or 12 months.“
2. **We agree that this data was missing and should be added. As requested, we now provide additional quantitative data on the density and morphology of spines. These points are highlighted in the revised manuscript. The data on morphological spine fractions are summarized in the revised Figures 3, the data on absolute densities of the morphological spine classes has been rearranged into supplementary figure S1 and S4, according to their appearance in the manuscript.**
3. “There is no data to show if the levels or distribution of alpha-synuclein changes within dendrites at early time points (3-4.5 months) and late (6 and 12 months). This would be important to correlate if the observed changes are caused by an increase of alpha-synuclein accumulation.”
3. **As we mentioned in our answer for point 1, we regret that in dendrites bright enough for spine analysis it is technically not possible to stain for additional antigens. We have however stained for and quantified the human α -syn expression in the cortices of the mice used in this study. We find no significant difference in the intensity of staining, suggesting that the cortical accumulation of α -syn is relatively uniform across our age groups. This data is, along with data on the presynaptic density, in the new supplementary figure S5.**
4. “Figure 4 shows a curious accumulation of pS129 synuclein signal and ubiquitin within Nissl-positive nuclei (?) of the neurons (Fig 4f-g). This is strange, as both signals should be cytoplasmic. Provided Nissl staining images do not provide a good discrimination between nucleus and cytoplasm. Also, ThT staining of the cortex shows an abundant staining in PFF injected animals. ThT is known to produce a high nonspecific staining; therefore PBS injected control animals should be stained as well.”
4. **We thank this referee for this justified criticism and provide new and additional images. We agree that Thioflavin S staining creates a significant nonspecific staining, which is also present in control brain slices. In PFF seeded brains however, there are large and bright accumulations with an obviously fibrillary shape, which are not present in control brains. There, stained structures are smaller and have diffuse, undefined shapes. We have adjusted our Figure 4 accordingly to better show these points.**
As for the signals for pS129 and Ubiquitin, the aggregates are indeed cytoplasmic, and sometimes appear wrapped around the nucleus, which appears darker than the cytoplasm in our Nissl staining. Because of the three dimensional structure of the aggregates, maximum projections of image stacks are shown in the figure, since it is difficult to appreciate their shape in an optical section. Therefore some of the depicted signal is actually above or below the nucleus in the z dimension. We have now clarified this in the figure legend and here provide two optical sections of the same images to support our point.

5. “Two models presented are not really comparable. There is an upregulation of human transgenic alpha-synuclein in PDGF- α -synuclein mice, while in PFF injected animals the main effect comes from aggregation of endogenous murine alpha-synuclein. Do authors believe that these changes in the spine morphology are caused by alpha-synuclein overexpression within dendrites? This should not be the case in PFF model, as endogenous levels of alpha-synuclein are not changed. How far PFF's spread in this mouse model? From Fig.S2h, i it seems that PFF did not spread at all. Alternatively, there is no aggregation or phosphorylation of human (or murine) alpha-synuclein reported in PDGF model.”
5. **We thank this referee for his/her suggestions. For the PFF model, we have added a new section to the supplementary figure 2 (S2 D), illustrating the spread of α -syn aggregates 5 months after seeding with PFFs throughout different brain regions. In the PDGF model, mostly soluble α -syn strongly accumulates within neurons and presynapses (as shown in figure 1 and S6) and has been described to also be phosphorylated (Amschl *et al*, 2013). We regard the notion of α -syn expression within the dendrite itself as a possibility, but are rather supportive of the hypothesis that a change in the network activity and presynaptic function leads to excitotoxicity and ultimately postsynaptic loss. Although at the moment we can only discuss this, we are performing experiments on this issue, which will be subject of a future paper.**
6. “While authors speculate on possible mechanisms of such detrimental effect on dendritic morphology, it has to be clear whether they are talking about alteration in the conformation of endogenous alpha-synuclein (therefore loss of normal function), or overexpression of human alpha-synuclein (therefore gain of pathological function). The discussion part would benefit from more in depth analysis of these situations.”
6. **Thank you for this relevant criticism. We agree that from our imaging data, we cannot exactly pinpoint whether a toxic-gain-of function due to the increase in α -syn dosage or a loss-of-function due to the sequestration of functional protein into aggregates is more relevant for each model. It is likely that both cases would lead to a disruption in α -syn homeostasis, which ultimately would affect spine structure and dynamics. At this stage, we can only discuss the two mechanisms, as it clearly calls for further investigation. We have included and marked these points in the discussion of our revised manuscript.**

Reviewer #3

We thank this referee for his/her highly encouraging and positive remarks and greatly appreciate his/her interest in our study.

Specific points:

1. "Page 3 "in synucleinopathies like Parkinson's Disease or dementia with Lewy Bodies, progressive neurodegeneration due to misfolding and intracellular aggregation of the synaptic protein alpha-synuclein " Alpha-synuclein aggregation is believed to contribute to neurodegeneration, but the direct cause(s) of neurodegeneration are still unclear (as written in the reviews cited here). Please, correct the sentence in accordance."
 1. **Thank you for this relevant suggestion. The revised manuscript now states that misfolded and aggregated α -syn is *linked* to neurodegeneration.**
2. Page 6: The authors mention a brief sonication of the PFFs before use. Please, describe the sonication protocol.
 2. **This is relevant information, which was neglected in our previous submission. We now included the requested details into the methods part of the revised manuscript.**
3. Please cite and discuss additional literature exploring density of dendritic spines in alpha-synuclein transgenic models (for example, Finkelstein et al. 2016; and Oaks et al. 2013)
 3. **In our revised manuscript, we have included further discussion of the literature on α -syn transgenic and other models. These changes are highlighted in the discussion.**
4. It would be very interesting to know about the functional consequences of the dendritic alterations observed (electrophysiology recordings), but considering the amount of work, it could be the subject of another article.
 4. **We thank this referee for his/her valid suggestion. We agree that the functional alterations in our models are very interesting to study. We are performing such experiments in our laboratory, but at this stage have no more than preliminary results. We would like to include our data in one of our future articles.**
5. Supplementary figures are not cited in their order throughout the manuscript. Please, rearrange the Supplementary figures.
 5. **Thank you for your suggestion. We have rearranged our figures in accordance.**
6. Fig1. Error in the legend. Description of panel b and c are swapped.
 6. **Thank you for pointing this out. We have corrected this error.**
7. Fig3. Please, place panels a, b, c from left to right, and then panels d and e below them. It would be easier for the reader. Also, clarify the legend for each panel about the type of experiment was done (in vivo two-photon imaging or ex vivo confocal imaging). Also, panel a and d show a spine density range that is very different (0.3 vs 0.6 spines/ μ m) although the time is similar (3 months vs 2- 4.5 months). I suppose the difference is due to shrinkage of the tissue during the histology procedure? This could be mentioned in the results part.
 7. **As suggested, we have rearranged our figure 3 to make reading through it clearer.**
8. Figure 5 d. Please add in panel d the legend at the top of the related images "control" and "5mo PFF" (similar to panel b).
 8. **Thank you, we have corrected this obviously missing label.**

Additional references:

The following references, which are not included in the manuscript, additionally support our answers to the comments of the three referees.

Amschl D, Neddens J, Havas D, Flunkert S, Rabl R, Römer H, Rockenstein E, Masliah E, Windisch M & Hutter-Paier B (2013) Time course and progression of wild type α -Synuclein accumulation in a transgenic mouse model. *BMC Neurosci.* **14**: 1

Kahle PJ (2007) α -Synucleinopathy models and human neuropathology: similarities and differences. *Acta Neuropathol. (Berl.)* **115**: 87–95

Knott GW, Holtmaat A, Wilbrecht L, Welker E & Svoboda K (2006) Spine growth precedes synapse formation in the adult neocortex in vivo. *Nat. Neurosci.* **9**: 1117–1124

Liebscher S, Page RM, Käfer K, Winkler E, Quinn K, Goldbach E, Brigham EF, Quincy D, Basi GS, Schenk DB & others (2014) Chronic γ -secretase inhibition reduces amyloid plaque-associated instability of pre- and postsynaptic structures. *Mol. Psychiatry* **19**: 937–946

2nd Editorial Decision

22 February 2017

Thank you for the submission of your revised manuscript to EMBO Molecular Medicine. We have now received the enclosed reports from the referees that were asked to re-assess it. As you will see the reviewers are now supportive and I am pleased to inform you that we will be able to accept your manuscript pending final editorial amendments.

Please submit your revised manuscript within two weeks. I look forward to seeing a revised form of your manuscript as soon as possible.

***** Reviewer's comments *****

Referee #1 (Remarks):

The authors have addressed the main concerns raised from the previous review.

Referee #2 (Remarks):

I am happy with the corrections and author's responses to my comments.

Referee #3 (Comments on Novelty/Model System):

Very good work, and adequate analyses. The results are very convincing, the article is well written. The authors answered to all the reviewers' requests.

2nd Revision - authors' response

27 February 2017

We now resubmit our manuscript with the final editorial amendments as suggested in your last E-Mail and we hope you find them to your satisfaction. We have arranged our six supplementary figures as appendix and have rearranged our manuscript file according to your comments. We would like to point out that while indicating the exact p-values, we noticed that we had to update the number of *-symbols in Figures 3E and S1B (new graphs in the revised manuscript). This however changed none of the described results. We apologize for the previously not precise labelling and have corrected it.

Corresponding Author Name: Jochen Herms

Journal Submitted to: EMBO Mol Med

Manuscript Number: EMM-2016-07305